# PREGO: A Literature and Data-Mining Resource to Associate Microorganisms, Biological Processes, and Environment Types

**DOI:** 10.3390/microorganisms10020293

**Published:** 2022-01-26

**Authors:** Haris Zafeiropoulos, Savvas Paragkamian, Stelios Ninidakis, Georgios A. Pavlopoulos, Lars Juhl Jensen, Evangelos Pafilis

**Affiliations:** 1Department of Biology, University of Crete, Voutes University Campus, P.O. Box 2208, 70013 Heraklion, Crete, Greece; haris-zaf@hcmr.gr (H.Z.); s.paragkamian@hcmr.gr (S.P.); 2Institute of Marine Biology, Biotechnology and Aquaculture (IMBBC), Hellenic Centre for Marine Research (HCMR), Former U.S. Base of Gournes, P.O. Box 2214, 71003 Heraklion, Crete, Greece; sninidakis@hcmr.gr; 3Institute for Fundamental Biomedical Research, Biomedical Sciences Research Center “Alexander Fleming”, 16672 Vari, Greece; pavlopoulos@fleming.gr; 4Center for New Biotechnologies and Precision Medicine, School of Medicine, National and Kapodistrian University of Athens, 11527 Athens, Greece; 5Novo Nordisk Foundation Center for Protein Research, Faculty of Health and Medical Sciences, University of Copenhagen, 2200 Copenhagen, Denmark; lars.juhl.jensen@cpr.ku.dk

**Keywords:** text mining, microbiome data, literature-derived associations, comention statistics, biological processes

## Abstract

To elucidate ecosystem functioning, it is fundamental to recognize what processes occur in which environments (where) and which microorganisms carry them out (who). Here, we present PREGO, a one-stop-shop knowledge base providing such associations. PREGO combines text mining and data integration techniques to mine such what-where-who associations from data and metadata scattered in the scientific literature and in public omics repositories. Microorganisms, biological processes, and environment types are identified and mapped to ontology terms from established community resources. Analyses of comentions in text and co-occurrences in metagenomics data/metadata are performed to extract associations and a level of confidence is assigned to each of them thanks to a scoring scheme. The PREGO knowledge base contains associations for 364,508 microbial taxa, 1090 environmental types, 15,091 biological processes, and 7971 molecular functions with a total of almost 58 million associations. These associations are available through a web portal, an Application Programming Interface (API), and bulk download. By exploring environments and/or processes associated with each other or with microbes, PREGO aims to assist researchers in design and interpretation of experiments and their results. To demonstrate PREGO’s capabilities, a thorough presentation of its web interface is given along with a meta-analysis of experimental results from a lagoon-sediment study of sulfur-cycle related microbes.

## 1. Introduction

Microbes are omnipresent and impact global ecosystem functions [1] through their abundance [2], versatility [3], and interactions [4]. These facts have inspired microbiologists from diverse scientific fields to study their genotype and phenotype [5], their metabolism [6], and their interactions with the environment [7]. All this work has resulted in a wealth of knowledge available in the forms of literature and experimental data. Literature contains vast amounts of information in the free text form that overwhelms researchers. Advanced text mining methods [8] have been developed to assist this issue. Experimental data and their metadata require mining [9] as well for their integration, mostly through metagenomic mining from online repositories. Hence, the combination of this knowledge about microbial life (who), their metabolic functions (what), and the environment they influence (where) is an important step to study ecosystem function [10].

High Throughput Sequencing (HTS) has turned the page on microbial ecology studies [11]. Over the past 20 years, both the taxonomic and the functional profiles of microbial communities from both local and large-scale regions (e.g., Tara Oceans [12], Earth Microbiome [13]) are being accumulated at a higher and higher rate. Extreme environments, i.e., areas with high salinity, low pH, etc., are being studied, providing us with unprecedented insight [14]. Both amplicon and shotgun metagenomics studies have played a crucial part in this development. Latest technological breakthroughs, such as Metagenome-Assembled Genomes (MAGs) and Single Amplified Genomes (SAGs), are enhancing the assessment of the taxonomic and functional repertoire of microbiomes even further. However, the mass use of these technologies and their consequent data have led to a number of needs and challenges, with metadata curation being among the most crucial ones.

Standards-promoting communities, like Genomic Standards Consortium (GSC) (https://gensc.org/, accessed on 24 December 2021), their efforts, like Minimum Information about any (x) Sequence (MIxS) [15], and projects endorsing those, like National Microbiome Data Collaborative (NMDC) [16,17], offer guidelines and best-practices to assist the annotation of microbial ecology samples. Controlled vocabularies and ontologies contribute to these efforts as they describe each subject area with formal terms [18]. Environment types, for example, are described by the Environment Ontology (ENVO) [19]. Other key biological aspects that have been captured include molecular functions (Gene Ontology Molecular Function (GOmf) [20,21], Enzyme Commission nomenclature [22], etc.), and the pathways carrying out different biological processes (GO Biological Process (GObp), MetaCyc [23], etc.). These knowledge structures, along with taxonomic centralized resources like the National Center for Biotechnology Information (NCBI) Taxonomy [24] and LPSN (List of Prokaryotic names with Standing in Nomenclature) [25], provide the means for a standardized representation of, for example, environments, process-oriented terms, and microbial taxa, respectively. Global-scale public resources (like MGnify [26], JGI/IMG [27], MG-RAST [28]) combine some of the aforementioned resources to support the collection, analysis, and distribution of multiple types of HTS data (e.g., amplicon, metagenomics, metatranscriptomics, etc.).

Besides the data and the analyses *per se*, the related scientific literature stores valuable information in billions of text lines. PubMed [24] and PubMed Central (PMC) [29] are gateways to relationships among microbes (*who*), the environments they live in (*where*) and their associated processes and functions (*what*) hidden in text [30]. Text mining (on both literature and metadata) can serve the extraction of these relationships. Named Entity Recognition (NER) can, for example, locate organism names [31], ENVO and GO terms [32] mentioned in text and map them to their corresponding identifiers. Association statistics, like co-mention analysis, can subsequently suggest ranked association among such entities [33,34]. The new era of omics has been interwoven with data integration [35] by bringing together scattered and fragmented pieces of information.

The time is ripe for tools that integrate all this knowledge and henceforth assist researchers to tackle major challenges like climate change [36], sustainability [37], and synthetic ecology [38]. Many resources have emerged in this realm [39], each one serving a specific purpose, such as BacDive [40]. BacDive is a large-scale curated database with prokaryotic information about phenotypic, morphological, and metabolic information. Other resources like Microbe Directory [41], Web of Microbes (WoM) [42], and Microbial Interaction Network Database (MIND) [43] focus on microbial environmental conditions, metabolite interactions with microbes and microbe-microbe interactions, respectively. In addition, taking advantage of aforementioned resources, novel pipelines, e.g., [44], are emerging with the aim to explore the network associations of who (microbial taxa) is performing what (microbial processes) and where (environments) directly using graph theory [45]. These analyses and resources are important because microbiologists can enrich their data to explore hypotheses but also to identify potential gaps in knowledge regarding these associations [46].

Here, we present PREGO, a hypothesis generation web resource that is designed to be useful to microbiologists—in particular microbial ecologists and environmental microbiologists. Its specific aims include: (a) the gathering of source data, metadata, and literature followed by the extraction of microorganism, process, environment associations contained therein, (b) making such a mined knowledge base available to life sciences researchers via an easy to use and explore web portal. As such, PREGO can be useful also to system microbiologists and large-scale data analysts through bulk download and programming access. We document the principles, analysis methodology, and contents behind PREGO. Last but not least, we demonstrate PREGO’s capabilities for researcher-support related to the above through a case study involving sulfate-reducing microorganisms.

## 2. Materials and Methods

PREGO is a resource designed to assist molecular ecologists in acquiring a single point overview of *what-where-who* process–environment–organism associations. The system is comprised of two main parts: (a) a server that periodically harvests data and extracts process-environment-organism associations from the scientific literature, environmental samples, and genome annotation sequences (Figure 1, step 1 to 5) and (b) a web-based interface as well as an Application Programming Interface (API) that provides users and programmers with a friendly way to extract and navigate across the process–environment–organism associations (Figure 1, step 6).

### 2.1. Entity Types, Channels, and Associations

PREGO supports three entity types: *Process*, *Environment*, and *Organism*. For interoperability and consistency, an ontology or taxonomy is adopted for each type of entity. Processes are represented as Gene Ontology (GO) terms and are grouped either as Biological processes (GObp) or as Molecular functions (GOmf). In addition, Environments are represented by terms from the Environmental Ontology. Organisms are represented by the microbial NCBI Taxonomy Ids (Bacteria, Archaea, and unicellular eukaryotes). For the unicellular eukaryotes, a custom list was populated with the unicellular eukaryotic taxa using a curated list.

PREGO’s contents are mainly divided into three distinct channels of information based on data origin and format (Figure 1, step 1). The *Literature* channel exploits scientific publications, i.e., abstracts and full text open access scientific publications (Table 1 and Section 2.2). Through the *Annotated Genomes and Isolates* channel, PREGO retrieves genome annotations and their accompanying metadata (Table 1 and Section 2.3). Finally, the *Environmental Samples* channel supports the integration of metagenomic analyses from both amplicon and shotgun studies. These include taxonomic and functional profiles along with their corresponding metadata (Table 1, more details in Section 2.4).

In cases in which the retrieved data and metadata are in text form, they are standardized to the aforementioned identifiers and taxonomies using Named Entity Recognition (NER) tools, namely the EXTRACT tagger [32,47]. In cases where data contain KEGG Orthology terms and/or Uniref identifiers, they are mapped to the respective GOmf using the mapping files available from the UniProt (see Appendix B). Associations are extracted after the mapping and standardization of the entities from each resource (Figure 1, step 3).

The association extraction pipeline is distinct for each channel and resource because of differences in the data type origin (see *prego_gathering_data* in the Availability of Supporting Source Codes section). By the means of navigation, the large number of associations returned to the user require a type of sorting; ideally, one that ranks the most trustworthy associations at the top. For those reasons, each channel of PREGO has a dedicated scoring scheme bounded within the (0, 5] space for consistency. In Appendix D, the scoring scheme of each channel is elaborated.

### 2.2. Text Mining of Scientific Literature

PREGO implements a text mining methodology to extract associations of the aforementioned entities from literature. The origin of text mining is a corpus that comprises scientific abstracts and full text articles from MEDLINE^®^ and PubMed^®^ and PubMed Central^®^ Open Access Subset (PMC OA Subset) [48], respectively. The building and periodic update of the corpus is possible through the NCBI File Transfer Protocol (FTP) services. PREGO also has a dedicated text-mining dictionary (see Availability of Supporting Source Codes section) that contains the entities ids, names, synonyms, and neglected words (stop words). PREGO dictionary incorporates the ORGANISMS [31] and ENVIRONMENTS [49] evaluated dictionaries as well as the experimental dictionaries of Gene Ontology Biological Process and Molecular Function.

Text mining is subsequently performed on the corpus using the dictionary through the EXTRACT tagger [32,47]. The tagger recognizes the entities of the dictionary in each abstract and full text article and assigns their co-mentions with a score. The score is sensitive to the text structural level of co-mention; higher to lower scoring when co-mention appears in the same sentence, then, in the same paragraph, and lastly in the same article. All these are integrated and normalized to a single score for each association, as implemented in STRING 9.1 [34] (see Appendix D for more details). In addition, the tagger extracts each mention in every article to provide the origin of each association it extracts.

### 2.3. Annotated Genomes and Isolates

Annotated genomes and isolates comprise the most trustworthy data in PREGO’s knowledge base because they refer to a single species/strain and also have manually curated metadata. Among other data types, JGI-IMG [27,50] includes millions of genes from isolated genomes (*isolates*), SAGs and MAGs. Such annotations, along with their corresponding metadata, were collected using web-parsing technologies. Their metadata, describing their related environment/ecosystem, were tagged using the EXTRACT tagger to infer *organisms*—*environments* associations. The annotated KEGG terms were mapped to GOmf terms (see Appendix B). The GOmf terms were then used to extract *organisms*—*processes* associations.

The Struo pipeline [51] and its outcome when using the Genome Taxonomy DataBase (GTDB) (v.03-RS86) [52] was exploited to enrich *organisms*—*processes* associations. A set of 21,276 representative genomes, accompanied by UniRef50 annotations, was retrieved using the provided FTP server. The annotations were then mapped to GOmf terms (see Appendix B). Related GTDB genomes were mapped to their corresponding NCBI taxa (see Appendix B). All associations extracted from these resources were assigned arbitrarily a confidence level of four out of five. This score choice reflects the high-quality of these data and metadata.

In addition, BioProject data were integrated to PREGO using the NCBI FTP/e-utils services [48]. The BioProject ids that were integrated are the ones that have been assigned a PubMed abstract, a unicellular taxon, and Genome sequencing as data type. Then, using the text mining pipeline, associations were extracted connecting the assigned taxon with the rest of the entities that appear in the abstracts. This method resulted in associations that were assigned a confidence level of three (out of five) because of the combined method of curated data with text mining.

### 2.4. Environmental Samples

MGnify [26] and MG-RAST [28] repositories provide a great number of public metagenomic records. In the PREGO framework, both amplicon and shotgun metagenomic analyses are retrieved periodically along with their corresponding metadata. Data retrieval from these resources is possible from their Application Programming Interfaces (APIs). Marker gene analyses are retrieved and by measuring the co-occurrence of taxa present in the various environmental types (e.g., biomes, materials, features, etc.) *organisms*—*environments* associations are extracted. These associations emerge when a taxon is reported together with a certain environmental type, being mentioned in the metadata of a sample (*metadata based co-occurrence*). Similarly, analyses of metagenomic samples along with their corresponding metadata and annotations are also retrieved and *organisms*—*environments*, *organisms*—*processes* and *processes*—*environments* are extracted. The *processes*—*environments* associations are possible through co-occurrence of the functional annotations of metagenomes with the environmental metadata of the samples.

In all cases, the EXTRACT tagger is used on the microorganism names and the corresponding metadata of each sample to identify their identifiers (NCBI ids, ENVO terms, GOmf, GObp). All associations in this channel are scored based on the number of samples the entity of interest co-occurs with specific sample metadata (e.g., environmental type) or annotations (functional annotations or taxonomic annotations). The same scoring scheme was implemented across the channel resources (see Appendix D for more details), which ranks these associations with a value in the (0, 5] space.

### 2.5. Sequence Search

In the case of organisms, PREGO enables sequence-based queries, meaning a sequence (amplicon) can be used as an entry point like it was a taxon name. To this end, a custom database was built using a set of reference custom databases for four commonly used marker genes. For 16S and 18S rRNA, the SILVA database (v.138) [53] and the PR^2^ database (version_4.14.0) [54,55] were used. Cytochrome c oxidase I (COI) [56] is another commonly used marker gene; for this reason, Midori 2 (version GB243) [57] was integrated in PREGO’s custom database. Finally, for the Internal transcribed spacer (ITS), common in studies focusing on Fungi, the Unite (version 8.3, accessed 10.05.2021) [58] database was added.

### 2.6. Back-End Server and Front-End Implementation

PREGO is a multi-tier web-based application. It is hosted on a 64 GB RAM DELL R540, 20 core, Debian server. Custom API clients (written in Python) are responsible for retrieving the data and metadata from each source (Figure 1, step 2). These clients as well as the subsequent methodology (Figure 1, step 3 to 6) are updated in regular cycles using custom daemons (see Appendix C, Figure A1). The *mamba/blackmamba* web framework underlies communication to the Postgres association-holding database and the client-side communication. HTML 5, Ajax, JQuery, and custom Javascript enhance the user web experience. PREGO supports widely used browsers (e.g., Chrome, Firefox, Safari, Edge) in various operating systems, such as Windows 10, Linux (Ubuntu 18), and MacOS (10.12, 11).

## 3. Results

### 3.1. The PREGO Web Resource

Users can access the PREGO contents through its web User Interface (UI) (Figure 2 and Figure 3), its Application Programming Interface (API) (Figure 4), or bulk download of all associations (Appendix E). The User Interface comes with two search fields: a plain text search and a sequence search (Figure 2a). The latter is used when the user wants to search for a taxon sequence (see Section 2.5 for supported sequence databases). The plain text search supports three types of entry points; the user can search for a taxon name, e.g., *Methanosarcina mazei*, an environmental type, e.g., *lagoon*, or a biological process e.g., *methanogenesis*. In all entry points, PREGO returns an overview page consisting of tabs with associations of the entity of interest with the entities of the two other types (Figure 2b–d) as well as Documents and Downloads tabs (Figure 2e,f).

Regarding the association tabs, when a taxon is used as a query, PREGO returns an overview page consisting of tabs for environments, biological processes, and molecular functions. When an environmental type is used as input, PREGO returns the organisms that have been found to be related to it, as well as the Biological Processes observed in the given environment. Lastly, if a biological process is under study, PREGO returns a tab with the organisms along with another tab with the Environments related to the process. Notably, only the associations with scores higher than 0.5 are presented in the web platform and are sorted in descending order based on their score. The score is visualized with a five-star system (see Appendix D for the scoring scheme).

Every association tab contains three tables with associations derived from the PREGO channels (see Section 2) along with their supported evidence. The user can both search and scroll through these tables, which makes knowledge extraction easier in cases where, for example, Isolate data contain hundreds of associations. In the *Literature* channel, each association is supported by the scientific articles with text-mining identified co-mentions. When a user clicks on an association, a popup window appears. This window displays abstracts or excerpts of full text with the associated entities highlighted (Figure 3a). Additionally, the *Environmental Samples* and *Genome annotations and Isolates* channels support evidence for each association by providing links to more detailed information. In the former channel, when the users click on an association, they are redirected to pertinent sample pages of MGnify (Figure 3b). Similarly, the latter redirects users to JGI and NCBI genomes when the associations originated from JGI—IMG and Struo, respectively (Figure 3c).

The *Documents* tab includes a list of scientific publications where the queried entity is mentioned. Through the *Downloads* tab, users are able to get all of the PREGO associations found for their query, per entity type (e.g., all the environments found related to an organism) and per channel (e.g., all the Environments found related to an organism through the *Literature* channel). This data retrieval functionality is also available via the PREGO API (syntax described in Figure 4). Finally, all PREGO associations are available for bulk download from each channel (see Table A2).

### 3.2. PREGO in Action

To demonstrate PREGO’s potential, we present four different ways that PREGO can assist molecular ecologists. The demo focuses on the sulfate-reducing microorganisms (SRMs) as well as the processes and environments that relate to sulfate reduction. Through this demo, we highlight how the different channels may provide complementary insights regarding different taxonomic levels and different association types.

#### 3.2.1. Which Environments Are Related to a Taxon?

Based on Pavloudi et al. (2017) [59], several bacterial and archaeal SRM were found in lagoonal sediments, after amplifying and sequencing the dissimilatory sulfite reductase β-subunit (dsrB). Using PREGO for the case of Desulfobacteraceae, the family in which the majority of the observed OTUs of the study belonged to, several environmental types similar to lagoons were retrieved from both the *Literature* and the *Environmental samples* channels (Figure 3a,b). Moreover, most of them had a high z-score, such as “*sediment*”, “*sludge*”, and “*activated sludge*”. Several dissimilar environmental types were associated with Desulfobacteraceae, e.g., “oil reservoir” indicating them as potential environments where sulfate reduction takes place. However, the presence of taxa within that family in different environments, from “sea water” to “forest” and “Wastewater treatment plant”, may suggest that this family has ubiquitous representatives in diverse conditions.

Searching for *Desulfatiglans anilini* (https://prego.hcmr.gr/example1, accessed on 24 December 2021) at the species level, the most abundant species in Pavloudi et al. (2017) and, for *Desulfatiglans anilini* DSM 4660 strain (https://prego.hcmr.gr/example2, accessed on 24 December 2021), PREGO provides associations with the “*Anaerobic sediment*”, “*Marine oxygen minimum zone*”, and “*Anaerobic digester sludge*” terms. These associations further corroborate the relationship between the species and sulfate reduction. More specifically, the “*sulfur spring*” ENVO term was retrieved from the *Environmental samples* channel as well.

#### 3.2.2. Which Biological Processes and Molecular Functions Are Related to a Taxon?

According to Pavloudi et al. (2017), *Desulfatiglans anilini* plays an important role in sulfate reduction. The *Biological Processes* provided by PREGO’s Literature channel are the GO terms “*Sulfate reduction*”, “*Sulfide oxidation*”, and “*Sulfide ion homeostasis*”, which support this claim. In addition, the “*Denitrification pathway*” term was also retrieved. This is rather interesting as it is in line with what Pavloudi et al. (2017) discussed about the SRMs and their ability to use various electron acceptors, e.g., nitrate and nitrite.

Furthermore, PREGO’s *Molecular Function* tab provides more insight on this example. Several GO terms related to sulfate reduction (e.g., terms related to “*sulfite reductase*”) were associated with DSM 4660 strain and *Desulfatiglans anilini* species in multiple channels. Interestingly, in the case of the strain query, the *Annotated Genomes* channel returned many GO terms related to the nitrogen fixation, e.g., “*nitric oxide dioxygenase activity*”.

#### 3.2.3. Which Taxa Are Related to a Biological Process?

PREGO can be also used to report organisms that relate to a certain biological process. Searching for “*dissimilatory sulfate reduction*” associations with taxa (https://prego.hcmr.gr/example3, accessed on 24 December 2021) resulted in several taxa that were mentioned in the Pavloudi et al. (2017) study. For example, taxa such as Thermodesulfobacteria and *Thermodesulfovibrio* were found among the entries with the highest score (e.g.,) based on the *Literature* channel. The other two channels did not contain any associations. Using the “*Sulfate assimilation*” (https://prego.hcmr.gr/example4, accessed on 24 December 2021) as the biological process input, PREGO results showed several genera that were missing from PREGO results concerning the “*dissimilatory sulfate reduction*”. Hence, manual search of GObp terms that describe the actual biological process of interest is more insightful.

#### 3.2.4. Are There Any Associations between Environments and Biological Processes?

Are there other environmental types, except the lagoonal sediments, in which sulfate assimilation occurs? In that question, and in “*dissimilatory sulfate reduction*” (https://prego.hcmr.gr/example3, accessed on 24 December 2021) in particular, PREGO assigns the highest score to “*sediment*” while, among others, “*anoxic water*”, “*oil reservoir*”, “*mud volcano*”, and “*basalt*” are potentially associated with environments related to sulfate reduction.

Inversely, PREGO is insightful about occurring processes in a specific environmental type. For example, searching for the biological processes that take place in “*basalt*” (https://prego.hcmr.gr/example5, accessed on 24 December 2021), processes like “*Nitrogen fixation*” and “*Reactive nitrogen species metabolic process*” stand out. However, sulfate reduction is not among the associations. However, when asking for “*Mafic lava*” (https://prego.hcmr.gr/example6, accessed on 24 December 2021), both the “*nitrogen fixation*” and “*Sulfur compound metabolic process*” terms are returned. This highlights the suggestions of Pavloudi et al. (2017), regarding the potential use of various electron acceptors from the different strains present in different environmental types.

### 3.3. PREGO Contents

PREGO contains the literature, environmental samples, and genome annotations of the resources shown in Table 1. The extracted contents of these resources have resulted to a knowledge base with ~364 K distinct taxonomic groups (out of a pool of ~620 K Bacteria, Archaea, and microbial eukaryotes, based on NCBI Taxonomy) from which ~258 K are at the species level (Table 2). These taxa are associated with ~1 K Environment Ontology terms, ~15 K GObp terms, and with ~7.9 K GOmf terms. Combining the above, PREGO maintains a knowledge base of entities and associations between them that form a multipartite network with entities as nodes and scored associations between them as weighted links.

As shown in Figure 5, in its current version (December 2021), PREGO knowledge base covers 157 bacterial phyla (107 are Candidatus), 23 phyla from archaea (18 are Candidatus), and 22 unicellular eukaryotic phyla described in the NCBI Taxonomy database. The number of bacterial taxa present among the associations of each phylum ranges from the order of 10 s, as in the case of Candidatus Coatesbacteria, to hundreds of thousands, e.g., Actinobacteriae. The number of environmental types, found among the PREGO associations for each phylum, ranges from just a few to up to 1000. Similarly, the number of biological processes that have been related to the various phyla may range from less than a dozen, e.g., Yanofskybacteria to up to several thousands, e.g., Bacteroidetes. On the contrary, the number of molecular functions found to be related to taxa of each phylum is rather constant in all three domains.

## 4. Discussion

### 4.1. PREGO Contents

On its current version and according to the NCBI Taxonomy that it is based on, PREGO manages to cover a great range of microbial taxa, as most (if not all phyla) are present in the knowledge base (Figure 5). The different number of organisms’ entities per phylum highlights the diverse number of the members of the various phyla. On the contrary, the similar number of molecular functions in all cases indicates the robustness of the main metabolic processes required for life. With respect to biological processes, their number per phylum varies to some extent, especially for the case of Bacteria and Archaea. That could be observed as, in many cases, phyla that have been recently described using molecular techniques have not been studied extensively yet, e.g., Candidatus Delongbacteria. As expected, the number of environmental types that have been associated with members of each phylum varies, as a phylum may be universally present, while others could be strongly niche-specific (e.g., Hydrothermarchaeota).

Because of its three different channels, PREGO manages to extract associations both in the species and higher taxonomic levels. The *Isolates* channel supports explicit associations at the species level (Table 3 and Appendix A). Interestingly, the number of such genomes seems to have reached a plateau for now, as PREGO-like platforms include the same order of magnitude. The *Literature* channel, on the other hand, promotes the extraction of associations at higher taxonomic levels (Table 3 and Appendix A). This also applies to *environment*—*organisms* associations derived from the Environmental Samples channel (Table 3 and Appendix A). Associations regarding *biological processes*, though, are strongly enhanced by the *Literature* channel and the massive increase of literature.

Additionally, the text mining methodology of the *Literature* channel has retrieved most of the entities present in PREGO knowledge base (Table 2). A significant contribution to the taxa with associations is due to the PMC OA processing by the text mining pipeline of the Literature channel. This is in-line with reports in other applications of text mining when using full text articles [60]. However, the resulting associations are suggestive because of the text mining nature, and therefore subject for further review by the users.

### 4.2. Related Tools’ Functionality and Content

There is an emerging *niche* for tools similar to PREGO to bring forward microbe associations and metadata. Table 4 summarizes the common and different features of BacDive, WoM, NMDC data portal, and PREGO. All of them commonly share the environmental associations and biological/metabolic processes with the microbes.

BacDive is a well-established platform with a focus on phenotype and cultivation information for about 100,000 prokaryotes, bacteria, and archaea. It has a high level of curation for most of its input types, like literature, internal databases, and personal collections. The NMDC data portal has published the scheme, the user interface, and a demonstrative collection of samples that will be populated later on. Standout features are the spatial visualization with coordinates and the detailed information of the samples, e.g., sequencing instruments and methodology. An alternative approach is facilitated by WoM, which aims to bind chemistry to microbes. An environment, in particular, is defined as the starting metabolite pool that is transformed by an organism. Another tool is The Microbe Directory that contains fully curated metadata for about 8000 microbes from all superkingdoms. This tool focuses on conditions of growth and on host taxa.

Complementary to these tools, PREGO contains associations of bacteria, archaea, and eukaryotes. Distinctive features are the associations of environments with processes/functions and the large-scale literature integration with text mining. Most importantly, most of the tools are complementary to each other with minimum overlap, an indication of the opportunities for further innovative synergies.

### 4.3. PREGO Next Steps

PREGO is a user-friendly association mining and sharing platform. Its modular web-architecture grants it the flexibility for further improvements in the aforementioned aspects, namely: source datasets, user interface, entity, and association scope expansion. Regarding datasets, additional data, such as transcriptomes from MGnify and other records annotated with metadata from studies in EuroPMC (https://ebi-metagenomics.github.io/blog/2021/11/17/Publication-Annotations/, accessed on 24 December 2021) [61], could be incorporated. Similarly, the NMDC data platform standards-compliant annotated records (https://data.microbiomedata.org/, accessed on 24 December 2021) could serve as an additional resource with its high-quality metadata [16,17]. Reciprocally, if requested, pertinent literature and association summaries could be programmatically offered to interested third parties.

Furthermore, the entity types supported by the PREGO system could be expanded. For example, GOmf terms could be upgraded as a search-entry point to the system. In addition, disease and tissue describing terms, already supported by the PREGO-underlying EXTRACT system [32], could enter the PREGO ecosystem of associated entities. From a statistics perspective, the calculation of a combined association score, when an association is reported by more than one channel of information, could be another feature to add.

The user interface can be enhanced to support multiple entity and/or sequence queries, instead of single ones. Sequences can be processed by taxonomy assignment pipelines (e.g., PEMA [62]) and be converted into searching PREGO for associations. In addition, network visualization tools, like Arena3D^web^ [63], could allow interactive browsing of associations through multi-layered graphs. Enrichment analyses, like those performed by OnTheFly^2.0^ [64] or Flame [65], can be incorporated. Omics data analysis pipelines, like MiBiOmics [66], environment associations with sequences using SeqEnv [67] and biogeochemical associations with metagenomic data with DiTing [68] could be enabled by comparing the associations pertinent to different groups of entities. The computationally intensive tasks of multiple queries, taxonomy assignments to sequences and enrichment analysis could be offered by our in-house High Performance Computing facility (https://hpc.hcmr.gr/, accessed on 24 December 2021) [69] in synergy with pertinent Research Infrastructures like ELIXIR (https://elixir-europe.org, accessed on 24 December 2021) and LifeWatch ERIC (https://www.lifewatch.eu/, accessed on 24 December 2021).

**Availability of Supporting Source Codes:** The PREGO software modules are available under BSD 2-Clause “Simplified” License. Scripts, where additional libraries have been used, are subject to their individual licenses. More information on each module can be found as listed below:prego_gathering_data https://github.com/lab42open-team/prego_gathering_dataprego_daemons https://github.com/lab42open-team/prego_daemonsprego_mappings https://github.com/lab42open-team/prego_mappingsprego_statistics https://github.com/lab42open-team/prego_statistics

Additional software and curated lists along with their individual license are:tagger https://github.com/larsjuhljensen/tagger, BSD 2-Clause “Simplified” Licensemamba https://github.com/larsjuhljensen/mamba, BSD 2-Clause “Simplified” Licensetagger dictionary https://download.jensenlab.org/ and there in: https://download.jensenlab.org/prego_dictionary.tar.gz, CC-BY 4.0 license.

## Figures and Tables

**Figure 1 microorganisms-10-00293-f001:**
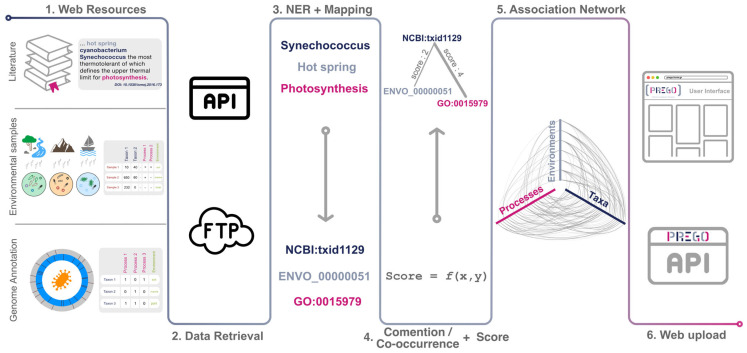
PREGO analysis methodology: PREGO periodically retrieves three distinct types of data from open access resources. Scientific text, environmental sample data, and genomic annotations are handled with respective methodologies in order to standardize their entities. Named Entity Recognition and Comention/Co-occurrence analysis is the common framework in order to build a weighted association network with nodes being the entity identifiers. Lastly, all these associations are available through a Web interface and an API. All these steps have been implemented in an autonomous way with regular cycles of updates (see Appendix C). Icons used from the Noun Project released under CC BY: Books by Shakeel Ch., Bacteria by Maxim Kulikov, ftp by DinosoftLab, Mountain by Diane, Ship on Sea by farra nugraha, River by Chanut is Industries.

**Figure 2 microorganisms-10-00293-f002:**
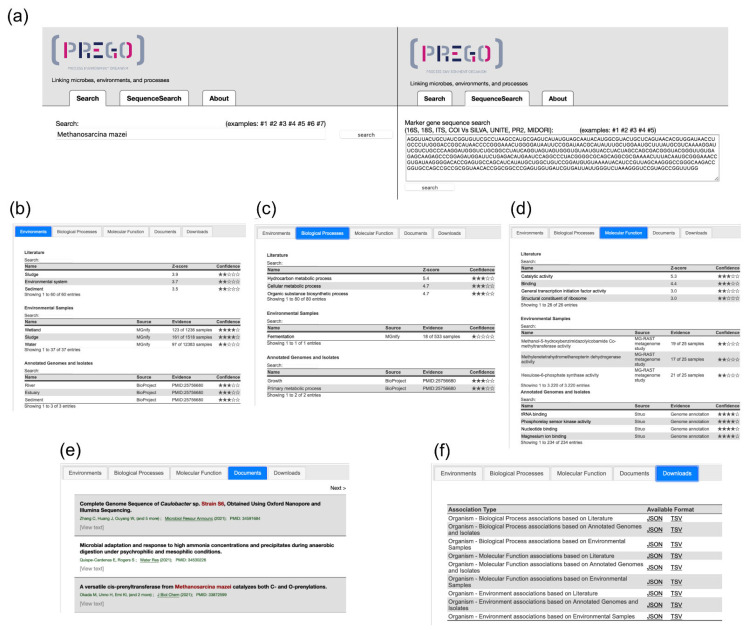
PREGO web user interface: (**a**) There are two search fields for user queries, plain text, and taxa sequences. (**b**–**d**) For the selected query, three association tabs are provided, each one presenting associations of the queried entity with the respective entities, Environments (**b**), Biological Process (**c**), and Molecular Function (**d**). Three channels of information distinguish the associations based on the original data. (**e**) The Documents tab presents the scientific articles that mention the queried entity with a highlighted color. (**f**) The Downloads tab provides the associations of each channel (when available) for download in JSON and TSV format.

**Figure 3 microorganisms-10-00293-f003:**
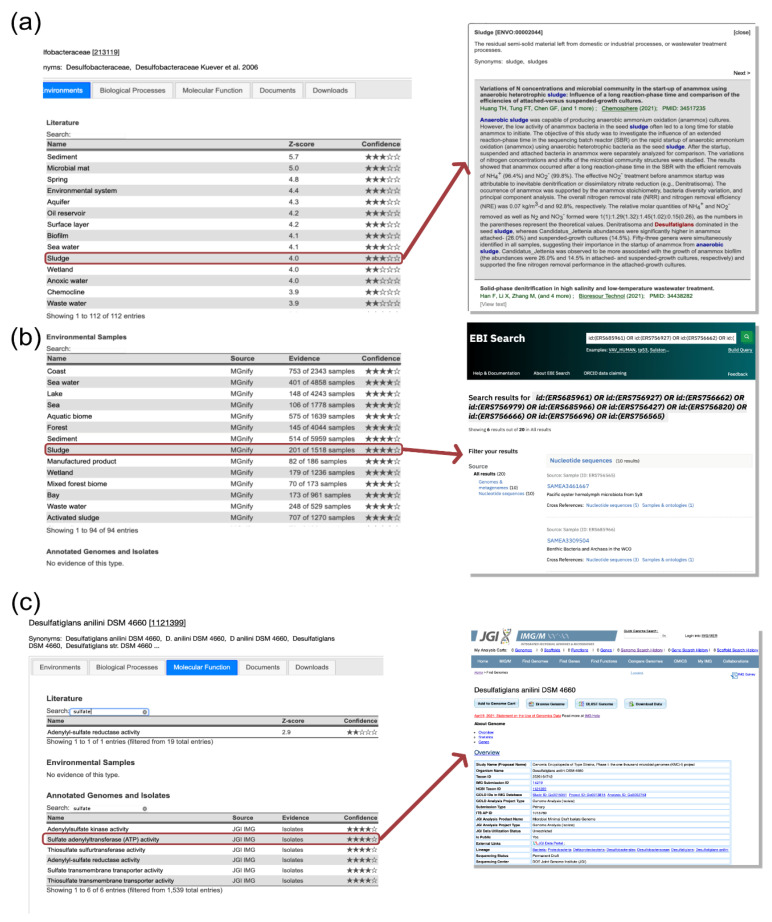
Each association is supported by original data: (**a**) *Literature* channel has a pop-up functionality that displays the scientific articles that support each specific association. The associated entities are highlighted in color. (**b**) *Environmental Samples* channel redirects to the samples that support the specific association (currently MGnify has this functionality). (**c**) *Annotated Genomes and Isolates* channel similarly redirects to the isolates ids that each association is based on (both Struo and JGI IMG are supported).

**Figure 4 microorganisms-10-00293-f004:**
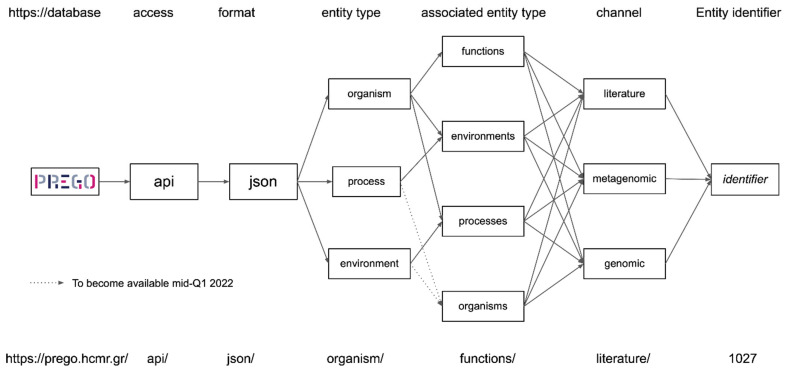
The PREGO API schema.

**Figure 5 microorganisms-10-00293-f005:**
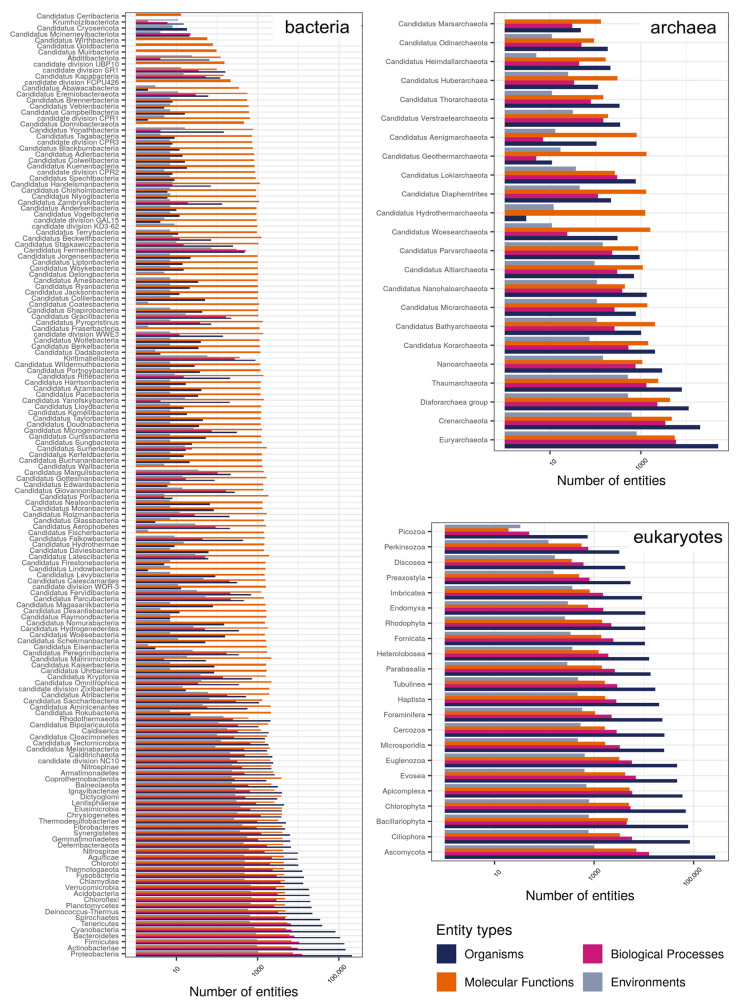
Summary of all the unique entities per phylum for each of the four entity types (in log10 scale) that appear in PREGO. Phyla are grouped based on their superkingdom (in log10 scale). Only phyla for which associations are available in the PREGO platform are mentioned.

**Table 1 microorganisms-10-00293-t001:** Source databases that are integrated in PREGO and the number of items retrieved. The Open Access subset of PubMed Central has a Creative Commons license available for commercial and noncommercial use. JGI has its own license, the same applies for BioProject, MEDLINE^®^, and PubMed^®^ as well.

Source	# Items	Data Type	Metadata	License
MEDLINE and PubMed	33 million	abstracts (text)	no	NLM Copyright
PubMed Central OA Subset	2.7 million	full article (text)	no	CC for Commercial, non-commercial
JGI IMG	9644	Isolates Annotated genomes	yes	JGI Data Policy
Struo	21,276	Annotated genomes	no	MIT, CC BY-SA 4.0
BioProject	18,752	Annotated genomes with abstracts (text)	yes	INSDC policy
MG-RAST	16,096	marker gene samples	yes	CC0
	7965	metagenomic samples	yes	CC0
MGnify	10,500	marker gene samples	yes	CC-BY, CC0

**Table 2 microorganisms-10-00293-t002:** The entities of PREGO after the NER and mapping of every source: Counts of distinct entities of Taxa, Environments (ENVO terms), Biological Processes (Gene Ontology Biological process), and Molecular Function (Gene Ontology Molecular Function).

Channel	Source	Taxonomy		Environments	Biological Processes	Molecular Function
Literature	MEDLINE PubMed—PMC OA	Strains	8929	1077	15,079	7318
		Species	240,377			
		Total	342,506			
Environmental samples	MG-RAST amplicon	Strains	1392	162	-	-
		Species	4324			
		Total	5859			
	MG-RAST metagenome	Strains	2522	258	-	3839
		Species	4406			
		Total	7157			
	MGnify amplicon	Strains	2	216	11	-
		Species	1471			
		Total	2955			
Annotated Genomes and Isolates	JGI IMGisolates	Strains	2398	241	-	3670
		Species	11,203			
		Total	13,849			
	STRUO	Strains	6	-	-	2789
		Species	19,289			
		Total	19,325			
	BioProject	Strains	5754	309	626	-
		Species	3373			
		Total	9393			
Total	All	Strains	12,840	1090	15,091	7971
		Species	258,352			
		Total	364,508			

**Table 3 microorganisms-10-00293-t003:** The associations between entities of PREGO after co-occurrence analysis: The supported entity types of associations are Environments—Biological Processes, Environments—Molecular Functions, Taxa—Environments, Taxa—Biological Processes, Taxa—Molecular Functions.

Channel	Source	Environments—Processes	Environments—Functions	Taxonomy	Taxa—Environments	Taxa—Processes	Taxa—Function
Literature	MEDLINE PubMed—PMC OA	883,997	422,579	Strains	69,968	590,630	384,079
				Species	778,877	3,501,635	1,961,920
				Total	1,669,608	7,969,310	4,613,827
Environmental samples	MG-RAST amplicon	-	-	Strains	13,645	-	-
				Species	39,007		
				Total	53,439		
	MG-RAST metagenome	-	620,846	Strains	262,106	-	8,626,328
				Species	103,913		10,715,548
				Total	372,301		19,950,096
	MGnify amplicon	-	-	Strains	18	-	
				Species	30,122	351	-
				Total	111,976	2097	
Annotated Genomes and Isolates	JGI IMGisolates	-	-	Strains	8229	-	3,461,693
				Species	42,141		13,216,559
				Total	50,888		16,821,850
	STRUO	-	-	Strains	-	-	1803
				Species			4,070,195
				Total			4,079,312
	BioProject	-	-	Strains	3263	7473	
				Species	4187	4294	
				Total	7641	12,169	
Total	All	883,997	1,043,425	Strains	357,229	598,103	12,473,903
				Species	998,247	3,506,280	29,964,222
				Total	2,265,853	7,983,576	45,465,085

**Table 4 microorganisms-10-00293-t004:** Feature comparison among platforms that facilitate knowledge discovery and integration of microbial data.

Functionality	BacDive	Web of Microbes	NMDC	PREGO
manual curation	high	high	intermediate	low
literature integration	limited	no	no	yes
environment—taxa associations	yes	yes	yes	yes
environment—process/function associations	no	no	no	yes
process/function—taxa associations	yes	yes	yes	yes
phenotypic data	yes	no	no	no
data origin	original, integration	original	original, integration	integration
spatial coordinates	yes	no	yes	no
application programming interface	yes	no	yes	yes
bulk download	limited	yes	yes	yes

## Data Availability

The PREGO-mined association-datasets are available under a CC BY 4.0 license. They can be accessed: a. using the Web interface https://prego.hcmr.gr/, b. via the PREGO API (see Figure 4), and c. via bulk download files (See Appendix E).

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
