# Peer review of "PREGO: A Literature and Data-Mining Resource to Associate Microorganisms, Biological Processes, and Environment Types"

_microorganisms, 2022, doi:10.3390/microorganisms10020293_

Round 1

Reviewer 1 Report

The authors presented a web interface that integrates various sources of associations between environments, organisms, biological processes and molecular functions by interrogating various databases. The tool is potentially useful for microbiology, ecology and environmental researchers.

I only have a few questions/comments, in no particular order as they appear in the text.

  1. In addition to the associations found in different channels within different tabs, the authors also report the confidence level, this is a very useful feature. However, I find the explanation of the scoring criteria used unclear, in particular in Environmental samples. There is inconsistent notation: see subscripts of "score". Further, the text in this section is very hard to follow, can the authors at least explain why it is necessary to use different fomulas for score in eq.(2) and (3)?
  2. Figure A2, in my view, should be in the main text rather than Appendix.
  3. Line 542, abbreviation should be OTU
  4. The whole manuscript could read better after careful English syntax checking.

Author Response

Please see the point-by-point response that has been attached as a PDF.
Author responses are highlighted in blue font.

Reviewer 2 Report

The authors introduced a microorganism knowledge base PREGO that is powered by established and curated data repositories with environmental and functional information and published literature. The authors developed a text extraction, transformation, and scoring pipeline that provides a rank for the environment, molecular function, and biological process associations with the taxa of interest.

Here are my comments:

  • The Literature channel exploits scientific publications, i.e.abstracts and full-text open access scientific publications (see Fig 1.A and Section 2.2). – No Figure 1A indicated, just Figure 1. Correspondingly, no Figure 1B and C indicators and caption are found.
  • see prego_gathering_data in the Availability of Supporting Source Codes section- can you add a description to the repository content?
  • Appendix C: The s1,1 is the number of instances 585 that two terms of X=1 and Y=1 are co-occurring, i.e the joint frequency. The marginals are 586 the s1 , . and s. ,1 for x and y respectively, which are the backgrounds for each entity type. – incorrect term indicator, table and equations have C term, not S
  • Appendix C, equation 1: what is the justification for each document entity weight? Equation 2: same question for a=0.6, why this coefficient was picked?
  • Appendix C, The final value of the score is the z-score divided by two and capped to a maximum of 4. – explain this transformation.
  • Annotated genomes and isolates comprise the most trustworthy data in PREGO’s knowledge base because they refer to a single species/strain and also have manually curated metadata. – is this status reflected in the downstream analysis? Are scores produced from this channel carry equal to less trustworthy channels' contribution to the knowledge base? If multiple channels report a specific taxonomy association, is it reflected in the score?
  • 3 Annotated Genomes and Isolates is hard to follow, given its most trustworthy status, maybe better to create a schematic pipeline with all listed components

I suggest improving the methods section and incorporating part of appendix C with the detailed scoring (for all data sources) into the main methods with the emphasis on Z-scores and confidence value retrieval as these metrics are key informers of the introduced knowledge base.

Figures S2 and S3 are not referenced in the main text. What purpose do they serve?

Abbreviations Typo: OUT Operational Taxonomic Unit

Author Response

(The authors gave the same response as above.)
